# SIMUNEO: Control and Monitoring System for Lung Ultrasound Examination and Treatment of Neonatal Pneumothorax and Thoracic Effusion

**DOI:** 10.3390/s23135966

**Published:** 2023-06-27

**Authors:** Adriana Rojas-García, Diego Moreno-Blanco, Martin Otero-Arteseros, Francisco Javier Rubio-Bolívar, Helena Peinado, Dolores Elorza-Fernández, Enrique J. Gómez, Manuel Quintana-Díaz, Patricia Sánchez-Gonzalez

**Affiliations:** 1Biomedical Engineering and Telemedicine Centre, ETSI Telecomunicación, Centro de Tecnología Biomédica, Universidad Politécnica de Madrid, 28040 Madrid, Spain; 2CEASEC—Fundación para la Investigación Biomédica del Hospital Universitario La Paz de Madrid, 28046 Madrid, Spain; 3Servicio de Neonatología, Hospital Universitario La Paz de Madrid, 28046 Madrid, Spain; 4Centro de Investigación Biomédica en Red en Bioingeniería, Biomateriales y Nanomedicina, 28029 Madrid, Spain; 5Servicio de Medicina Intensiva, Hospital Universitario La Paz de Madrid, 28046 Madrid, Spain

**Keywords:** simulation, simulator, neonate, lung ultrasound, pneumothorax, pleural effusion, web application, hardware

## Abstract

Training with real patients is a critical aspect of the learning and growth of doctors in training. However, this essential step in the educational process for clinicians can potentially compromise patient safety, as they may not be adequately prepared to handle real-life situations independently. Clinical simulators help to solve this problem by providing real-world scenarios in which the physicians can train and gain confidence by safely and repeatedly practicing different techniques. In addition, obtaining objective feedback allows subsequent debriefing by analysing the situation experienced and learning from other people’s mistakes. This article presents SIMUNEO, a neonatal simulator in which professionals are able to learn by practicing the management of lung ultrasound and the resolution of pneumothorax and thoracic effusions. The article also discusses in detail the hardware and software, the main components that compose the system, and the communication and implementation of these. The system was validated through both usability questionnaires filled out by neonatology residents as well as through follow-up sessions, improvement, and control of the system with specialists of the department. Results suggest that the environment is easy to use and could be used in clinical practice to improve the learning and training of students as well as the safety of patients.

## 1. Introduction

Pleural pathology in neonates requires specialised care in most cases. The most common conditions in neonates are pleural effusion (PE) and Pneumothorax (PTX).

PTX is defined as a collection of air between the parietal and visceral pleura. It is one of the most frequent emergencies in the newborn period, typically occurring in the first 2–3 days of life that still carries an additional burden of morbidity and mortality. Its prevalence varies depending on gestational age, ranging from 4.0% in early preterm, to 2.6% in moderate late preterm, to 6.7% in term neonates [1]. Many newborns with PTX have no symptoms, but some develop a life-threatening condition manifested by rapidly worsening symptoms of dyspnoea.

Traditionally, pneumothorax is diagnosed by chest X-ray (CTX), which remains the gold-standard imaging test. However, chest X-rays expose children to radiation and the diagnosis of small pneumothoraxes can be challenging [2,3]. Lung ultrasonography (LUS) is a relatively new technique which has revolutionised lung imaging diagnostics and is gaining popularity. Diagnostic markers for PTX can be detected using ultrasound, with a high degree of accuracy; LUS has been shown to have both high sensitivity and specificity for diagnosing pneumothorax in newborns [2,4].

PE consists of the accumulation of fluid between the two layers of the pleura. Chylothorax is the most frequent cause of pleural effusion in the newborn. Clinical manifestations are variable depending on the amount of fluid accumulated and the rate of its establishment. After confirming a PE via chest X-ray, a chest ultrasound is recommended to define its size and characteristics. Management involves treating the underlying cause, although if certain effusion characteristics are present or in case of cardiorespiratory compromise, drainage of the pleural space fluid is recommended.

Over the past 20 years, LUS has been successfully used to diagnose and monitor lung diseases of the newborn. In addition, it has played a crucial role in disease follow-up and facilitated lung disease management procedures [4,5,6]. Due to the sharp learning curve, ease of use, and minimisation of overall radiation exposure, the application of LUS has significantly increased or even completely replaced CTX in some neonatal intensive care units (NICUs) across the world [7].

In order to successfully treat both respiratory pathologies that cause air or fluid accumulation in the pleura, it is necessary to undertake adequate training that includes the practice of technical skills such as the mastery of diagnostic techniques and decompression of air or fluid present in the pleura. Despite the preparation and mastery of techniques to resolve these complications, the possibility of facing a real pneumothorax and/or pleural effusion is rare, and when it occurs, the patient’s life is in danger; so, it complicates the learning situation because of the high risk involved [8]. One of the main concerns when practicing medicine is undoubtedly patient safety, and it is necessary to consider multiple ethical and legal issues [9]. Compared to adult patients, these aspects are augmented in minor patients, and specifically in neonates, especially when faced with, e.g., learning the handling of instruments or mastering new techniques. In the latter case, the extreme fragility of their lives is evident, since most of them suffer from complicated pathologies; therefore, regarding education, teaching directly with the preterm newborn is extremely complicated for clinical instructors and clinical students [10].

Clinical simulation offers an ideal environment to achieve adequate clinical training and preserving patient safety by allowing the creation of real scenarios in which students can learn and practice the different situations that may arise in real life [11], thus gaining professional and personal skills and increasing their self-efficacy and self-confidence. Simulators are environments composed of two parts, hardware and software that reproduce the behaviour of a system or generate situations that help to reproduce reality, whose purpose can be educational, professional, or for leisure and entertainment [12].

In the field of neonatology, most simulators that recreate realistic scenarios for clinical training are high-fidelity simulators [13]. There are several simulators that allow the performance of pneumothorax decompressions and thoracic effusions, differing among them in the mode of feedback of the activity that they present. Micro-Preemie Nasco—LF01280 [14], PEDI^®^ Recién Nacido S320 [15], and TruBaby X [16] are some examples of simulators that present feedback based on subjective metrics. Subjective metrics are understood as those that are visual and difficult to quantify. SUPER TORY S2220 [17], STAT Baby avanzado Nasco—101-8010 [14], NENASIM [18], CAE BabySim [19], CAE Luna [20], and LuSi [21] are some examples of simulators that are developed in conjunction with software, which require the correct operation and return results that can be evaluated through objective metrics that are established in the interface itself. Objective metrics are understood to be those that are quantifiable and measurable. On the other hand, BABYWORKS [22], Echocom|Neo [23], POCUSNEO [24], and display U/S [25] are examples of simulators that allow the performance of neonatal lung ultrasound.

Table 1 displays two sets of information: the aspects to be covered by the system under development and the simulators mentioned above. This arrangement enables a comprehensive comparison of the characteristics addressed by each simulator. Out of the 13 simulators mentioned, 8 primarily focus on treating respiratory issues, while the remaining 5 concentrate on exploration and diagnosis using lung ultrasound. Among the eight simulators dealing with decompression, six allow for both air and water decompression, while the remaining two solely facilitate pneumothorax treatment. Concerning the software control of the simulator, 10 out of the 13 simulators possess an associated control interface.

Upon analysing the simulators presented in Table 1, it becomes evident that none of them fulfil all of the proposed areas. This paper introduces SIMUNEO (SIMUlador NEOnatal), a clinical simulator designed for neonatology scenarios. Professionals can enhance their skills by practicing the management of lung ultrasound, as well as resolving pneumothorax and thoracic effusions. Additionally, SIMUNEO incorporates control software, enabling the creation of clinical cases for resolution. It provides feedback through visual cues during the decompression of pneumothorax or effusion caused by water or air leakage. The associated interface guides the entire exercise, offering objective and measurable assessment metrics at crucial stages of the exercise.

## 2. Materials and Methods

SIMUNEO was designed by a multidisciplinary team made up of professionals from Hospital Universitario La Paz (HULP), Madrid, Spain, and the Biomedical Engineering and Telemedicine Centre (GBT) of Universidad Politécnica de Madrid (UPM), Madrid, Spain. This simulator is composed of two main parts: hardware, which includes the manikin and all of the electronic sensors and components to collect the data and the feedback for users, and the software, which controls the whole system and allows for data preservation, user management, and case creation. Figure 1 shows a schematic of SIMUNEO.

### 2.1. Hardware Design

The simulator hardware comprises the physical components of the system. This portion can be further divided into two distinct parts based on their purpose: the anatomical structure and the electronics. The complete anatomical structure, which houses a portion of the electronics, is covered with a silicone material that mimics human skin. This realistic simulation is essential for providing a lifelike experience during the training exercises.

#### 2.1.1. Anatomical Structure

The main goal of the anatomic component is to provide a realistic environment for clinical students. To achieve this goal, the design of this component focused on creating realistic feedback to the students, specifically in the sensations of punction and the realism of the manikin. This component was created from scratch starting from a CT scan of the ribs of a preterm newborn to create the thorax, which serves as a support for the rest of the components. The process of creating this anatomical component included the reconstruction of the sternum and up to a total of seven ribs, as well as the addition of two perpendicular supports that act as the base and central pillar of the thorax.

In addition, three boxes are placed in the pneumothorax and effusion puncture areas of the anatomical structure [26]. The design and distribution of the boxes were based on the pathology, the puncture technique that resolves it, space, and electronic issues. The design of these boxes is shown in Figure 2, with the only difference between the boxes being the depth, which is adjusted based on the drainage applied.

In the design of the boxes that make up the internal architecture of the system hardware, it was considered that the space should host a pressure sensor inside, which is positioned at its base covered by a silicone replacement that simulates the internal musculature and the rupture of the pleura before the passage of the needle. Likewise, the design includes an exit to the sensor legs so as not to damage its own and associated electronics. Another noteworthy detail is the design of the hollow sliding cover. This design enables the needle to be inserted through the central hole, which leads to the electronic activation mechanism when it comes into contact with the pressure sensor. Additionally, it allows for the attachment of the ribs corresponding to the intercostal space, ensuring a secure fit of the box and facilitating the final assembly process.

The base and support of the thorax are designed for hosting all of the components, ensuring isolation among them, and guaranteeing the usability and maintainability of the system. Figure 3 shows this structure which is designed as two separate spaces that are required to be isolated from each other. One of them hosts the electronics while the other contains the water pump in its respective container used to simulate the outflow of liquid in the pleural effusion. The structure also includes a mobile part which contains the elements that require replacement, like batteries.

The fixed box is divided into two spaces of equal section based on the measures of the printed circuit board (PCB), which is enclosed in the first section. The second section is a drawer for the mobile box and two safety stops are placed near the rear wall in order to prevent any contact between the batteries and the water pump of the first section. All of the measures of the external structure were adjusted for the correct stability of the system and the correct performance of the techniques.

#### 2.1.2. Electronic Component

Two circuits were designed to carry out each simulation: the lung ultrasound scan and the decompression of the pneumothorax or pleural effusion. Figure 4 shows the circuit associated with the ultrasound simulation composed of six magnetic field reed sensors. This first circuit divides the thorax into 6 zones, with these being the top, bottom, and both left and right sides of the thorax. Each zone has a reed sensor associated with it. Because the ultrasound probe contains a neodymium magnet embedded inside it, the sensors are activated when the probe passes by them. Thus, the probed zone is identified by activating one single sensor, without any overlapping among them.

Figure 5 shows the designed circuit for decompressions. Its main components are a pressure sensor and a pump. The decompression of the pneumothorax and effusion circuit detects a certain threshold of the pressure sensor so that the electrical current passes through the transistor, amplifying and allowing the opening of the relay gate that allows the activation of the pump for a certain time. It should be noted that the circuit depicted in Figure 4 is repeated a total of three times, one for each pressure sensor associated with either an air pump (pneumothorax) or a water pump (pleural effusion). The system is controlled by an Arduino MKR WIFI 1010 board and is activated by a switch located on the cover of the external structure.

### 2.2. Software Design

A management application was created that controls SIMUNEO and allows for data storage, user management, and case creation.

#### 2.2.1. Actors and Use Cases

Two main actors were identified: teacher and student. The teacher role was designed for non-resident physicians, specifically, clinicians and specialists of neonatology. Teachers can access their profile in which they can perform various actions, such as uploading new cases, managing these cases, and assigning them to the students, or visualising the results of each student and thus knowing the learning curve and improvement with the respective simulations. The student role was designed for medical students and resident physicians, either to learn the technique of lung ultrasound and decompression of pneumothorax and effusion from scratch in a real environment that ensures patient safety, allowing them to make mistakes as many times as necessary, or to train in these techniques at specific times during their training, as desired. Students are supervised by an assigned teacher who oversees their progress. They can perform simulation cases assigned by their teacher, practice the pneumothorax and effusion technique on their own, or review their evaluations and track their progression.

#### 2.2.2. System Architecture

The web application is based on the model–view–controller (MVC), which divides the application into the NoSQL database in MongoDB; a view composed by the user web interface in Angular; and finally, a controller based on the REST API in NodeJs. The connection to the hardware is established through the REST API by creating a UDP socket to provide a data transport service between the WIFI module containing the microcontroller of the hardware part and the web interface.

The database used is a data storage database that follows a non-relational schema composed of four entities, users, cases, clips, and evaluations, each containing several attributes that are generated with the use of the application. The user entity stores all of the data of the users who make use of the web application. These data are collected once the registration of a new user is completed and are later consulted during the login or when listing the students of each teacher. The second of the entities mentioned above, cases, store all of the data generated when the teacher uploads a case, completing the clinical information of the pathology requested by the platform. In relation to this last entity, a new one appears, video clips, since when the case is being uploaded it is necessary to assign a lung ultrasound video clip associated with each chest division zone detailed above. Thus, when the case is resolved in the simulation, the assigned video clip is played when the reed sensor corresponding to the zone in question is activated. Finally, the evaluation entity stores all of the data collected once the assigned case has been performed.

The interaction between the different components of the web interface and the screens and corresponding information were designed using state diagrams. The state diagram of the web interface that determines the use of the application is shown in Figure 6. It defines the functions that each role described above, teacher and student, can perform. The student has two possible ways to train with the simulator. They can either solve a complete clinical case that will be evaluated by the assigned tutor or train the decompression techniques on their own. The resolution of the clinical case is a complete ultrasound examination to identify the events observed and make a diagnosis. In case the diagnosis is correct, and it is a case of pneumothorax or pleural effusion, treatment will proceed.

### 2.3. Implementation and Communication

The connection between the hardware and software components of the developed system is performed by a UDP socket. This connection is established through the WIFI module of the microcontroller. The messages that are sent following the UDP protocol in response to the commands of the server depend on the specific part of the simulation currently performed by the user. These commands are created according to the part of the simulation in which the user is, dividing them into ultrasound simulation, pneumothorax decompression, effusion decompression, and end of the simulation. In this way, it is possible to obtain feedback on each of them and thus be able to know the results and provide a complete evaluation to the student, as well as to allow for the subsequent monitoring of their results.

### 2.4. Validation Design

For the initial presentation of the SIMUNEO prototype, a validation process was proposed, involving one of the collaborating entities of the project, the Hospital Universitario de La Paz. The neonatology department of the hospital was actively involved in the project, and it was suggested to conduct the validation with members of this department. As SIMUNEO aims to serve as a training tool in the clinical setting, the first validation of the system involved five residents from the department. The reason for selecting residents as the initial group for validation in the medical field was because they were currently in their training period. This enabled them to effectively compare the traditional learning method with SIMUNEO, allowing for a more objective evaluation of the system’s realism and its similarities to their typical training. These residents, from different years of the neonatology service at Hospital Universitario La Paz in Madrid, were recruited through the specialists of the department and willingly chose to participate in the validation process. As the recruitment took place within the neonatology department, the validation was also conducted within this specific department of the hospital.

The validation consisted of twelve statements that the residents needed to answer after completing a full simulation exercise. Additionally, short interviews were conducted with the residents to gather their personal experiences and perspectives on the simulator. The validation task began with the simulation of ultrasound diagnosis. Users needed to evaluate the neonatal thorax, exploring 6 zones, and provide a diagnosis.

Once the ultrasound simulation had finished, depending on the diagnosis provided, the exercise continued with the resolution of a pneumothorax or pleural effusion. Users needed to perform the corresponding technique and the performance metrics were registered and evaluated. Therefore, the events as they occurred, being either a pneumothorax or pleural effusion, are briefly summarised, justifying the chosen design:Insertion of the needle or catheter.Needle or catheter overcomes the resistance provided by the silicone and touches the sensor located at the base of the box, exceeding a certain pressure threshold.The threshold is exceeded, and the pump mechanism is activated.The contents of the pumps travel through the tubes and are expelled to the outside.

The users were required to provide their answers to the statements using a Likert scale [27], ranging from 1 to 5. A score of 1 indicates total disagreement, while a score of 5 means complete agreement with the statement. The validation statements were designed to assess the system’s usability and complexity. To achieve this, various statements were included to evaluate the ease of system use, encompassing the web interface and the performance of techniques on the manikin. Additionally, the user’s autonomy when operating the simulator without a need for specialist supervision was examined. Feedback was also sought out regarding the realism of the system, both visually and in terms of the sensory experience during ultrasound scans and the simulated decompression of air or water passing through the simulated skin. Lastly, a statement was included to gauge the ease of battery replacement and water recharge.

## 3. Results

The anatomical component of SIMUNEO is shown in Figure 7. In Figure 7A, the 3D model of the segmented thoracic skeleton can be observed. On the other hand, Figure 7B shows the 3D printed version of the model, which includes an added support for the rib cage as well as a base, and which is scaled appropriately to the needs of the healthcare professionals.

The electronics are integrated into the different structures developed (rib cage, internal and external boxes) with the aim of achieving a robust, manipulable, and safe system. The assembly and the fusion of all of the anatomical components were carried out following a bottom-up component implementation, i.e., from the external structure to the internal structure and the thorax.

The precise distribution of the elements within the internal space is crucial, as it significantly influences the simulation results. However, placing these elements can be extremely complex due to the limited available space. In this distribution and placement of the internal elements, it is necessary to take into account the precise placement of the magnetic field sensors to be able to divide the thorax into six zones. These six zones must not overlap, in order to be able to observe their corresponding video clip correctly once their associated sensor is activated by means of an ultrasound probe, which has an integrated neodymium magnet. Figure 8 shows the final composition of the hardware part of the simulator.

Figure 9 and Figure 10 show the performance of a lung ultrasound on the simulator as well as the complementary screen that the student observes while performing the ultrasound. In addition to the ultrasound images, the series of events that the student must choose to make a diagnosis of the exercise being performed is shown.

Students can also supplement their training and prepare the tasks using the training mode provided by the platform. This mode as well as the performance of a thoracic effusion decompression can be seen in Figure 11 and Figure 12.

Table 2 gathers the answers obtained from the medical residents of the neonatology department who participated in the validation process. As can be observed, the higher scores were obtained in the utility of the simulator, whereas the lower scores appeared in the usability questions. Despite this, all of the answers are positive (3 or greater). 

In addition to completing the validation questionnaires, the residents were also interviewed to gather information about their personal experience using the simulator. All five residents mentioned that they had prior experience with simulation exercises for adults, but had only been able to try a neonatal simulator, which they found quite challenging to use. They all agreed that they had never performed a neonatal lung ultrasound simulation task. Furthermore, four out of the five residents stated that they had never had the opportunity to practice pneumothorax or effusion procedures in neonates. They expressed that working on these techniques with the simulator would be extremely beneficial in building their confidence and experience when they eventually would have to perform real decompressions. Another point of unanimous agreement was the user-friendliness of the SIMUNEO interface. Overall, the residents’ responses in the interviews were optimistic about this initial version of the system, and they expressed encouragement for its completion and improvement to enhance their training.

## 4. Discussion

SIMUNEO is a clinical training simulator that allows for the safe training of lung ultrasound examination, as well as pneumothorax and pleural effusion decompression techniques. The system collects data from the simulation exercise and provides an objective evaluation for educational follow-up.

The resulting system is a combination of hardware and software that allows for the creation of a simulation scenario. Both parts communicate with each other and are dependent on each other. The sensorization of the physical part of SIMUNEO allows the electronic components to communicate with the system software. The global anatomical design that allows for the integration of the electronics, together with the communication of both parts via Wi-Fi, makes SIMUNEO a robust and portable system.

The validation results obtained from the questionnaires completed by the five neonatology residents demonstrate the potential of SIMUNEO as a valuable tool in the educational training of doctors. As mentioned earlier, the validation was designed to gather initial feedback on the system, focusing specifically on its usability and complexity. The average scores for questions related to ease of use, the need for additional knowledge or professional assistance, and confidence levels when using the simulator were consistently positive. These results indicate that SIMUNEO strives to be an intuitive, accessible, useful, and effective system.

Regarding the materials used to simulate the skin, specifically the silicone employed, the results reveal the achievement of a realistic physical and visual appearance, as well as a sensation similar to that experienced during decompression. The average score for the statement concerning battery replacement and water recharging exceeds the accepted threshold.

In terms of the scores obtained upon completion of the validation questionnaires, adhering to the parameters established by the Likert scale, the lowest average response score stands at 77.8%. This percentage confirms the positive reception of the simulator by the neonatology residents at Hospital La Paz in Madrid. This initial feedback on the first SIMUNEO prototype suggests that the simulator holds promise as a valuable educational tool in the clinical setting. However, further refinements are necessary before implementing it within the hospital environment.

Overall, the residents’ perception of the system is positive, as they view SIMUNEO as a beneficial tool that allows them to train, make mistakes, learn from them, and enhance their techniques and self-confidence.

The results of this first validation show that SIMUNEO covers all of the needs set out in the state of the art, as described in the Introduction section. It should be recalled that none of the commercial simulators that exist to date have managed to cover all of the requirements set out. Moreover, SIMUNEO manages to match more complex simulators such as LuSi, CAE Luna, or POCUSNEO, which provide feedback to the user thanks to objective, quantifiable, and measurable metrics. In addition to these metrics, SIMUNEO also provides visual metrics such as air or liquid outflow that contribute to the realism of the overall system. One of the points that makes SIMUNEO stand out from all of the simulators mentioned in Table 1 is that it manages to encompass lung ultrasound, pneumothorax, and pleural effusion techniques in a single system.

However, this simulator has some limitations. The first of these is related to the tightness of the anatomical structure. As the airtightness of the anatomical components cannot be guaranteed with total certainty, an alternative solution has been proposed to expel air and liquids through a separate channel. This alternative ensures the safety of the total system, and that the sensation experienced when passing through the different layers of the simulated skin prevails.

Thanks to the feedback obtained and the tests carried out, it has been detected that the main limitations of SIMUNEO are related to the design of the anatomical components of the system. Since one of the lowest scores obtained in the validation is related to component replacements, work will be undertaken on a redesign of the anatomical structure. In addition to the optimisation of the spaces in each of the structures that make up the hardware, improvements related to silicone replacements and the sealing of the hardware components are also being considered. Puncture sites could be composed of replaceable silicone patches in case of damage or wear. A complete sealing of the system would be advantageous both to provide greater robustness and allow for more movements, without a risk of damage. This would also ensure that air and liquid remain compressed within the internal structures, so that they exit through the puncture needle, without the risk of water leakage after several uses that could compromise the rest of the components. On the other hand, to improve the overall integration of the system, work will be undertaken on optimising the sensor circuitry as well as its communication with the software part and on the interface. Ultrasound scanning could be improved by inserting a mesh of sensors containing a greater number of sensors, which would increase precision and decrease error, as well as allowing a greater number of scanning positions that would increase the realism of this task.

At the software level, an improvement in the web interface with additional functionalities to optimise the simulator’s data management is envisaged. The data transfer between both systems, software and hardware, must also be reviewed to avoid information overload and the collapse of the Arduino’s memory. From the point of view of the electronics, further study of the voltage of the electronic components and the compatibility between them has to be carried out. Preserving communications between electronic components will increase the lifetime of SIMUNEO by optimising the system consumption. Also, in order to collaborate with the sustainability of the environment, the replacement of the batteries with rechargeable batteries is proposed. Finally, it is proposed to carry out an exhaustive validation of the system in the future with different profiles of healthcare professionals, in order to obtain more feedback about SIMUNEO and thus be able to contrast its functionality more objectively.

Despite the positive outcomes observed in this initial validation, it is important to exercise caution in interpreting them due to the small sample size, which may not be fully representative. Therefore, it is essential to conduct additional validations with a larger number of participants to further substantiate these findings.

## 5. Conclusions

The development process for SIMUNEO was presented, including the system hardware and software design and development, the communication between both parts, and the final implementation. In order to validate this system, a validation process was performed with five neonatology residents from Hospital La Paz de Madrid. After completing a simulation task, a questionnaire to measure their impressions of the simulation experience was sent to the participants, and the outcomes were analysed. The obtained outcomes show that SIMUNEO holds great promise as a valuable solution in the educational environment of hospitals. It serves as a tool that facilitates learning, the practice of techniques, and the development of confidence via its application. However, given the limited sample size in the validation, it is important to interpret these results with caution, and additional validations are necessary to validate and confirm these findings.

SIMUNEO addresses a significant challenge in the education and clinical training of trainee healthcare professionals. It serves to enhance patient safety by providing an alternative to direct practice on real patients. Through realistic scenario simulations, SIMUNEO enables trainees to independently navigate and resolve situations effectively. The simulator allows users to make mistakes, encouraging them to confront and learn from these errors, ultimately preventing their repetition. By incorporating this approach, healthcare professionals are better equipped to handle unexpected situations with confidence and preparedness. This not only reduces the likelihood of errors but also indirectly enhances patient safety.

## Figures and Tables

**Figure 1 sensors-23-05966-f001:**
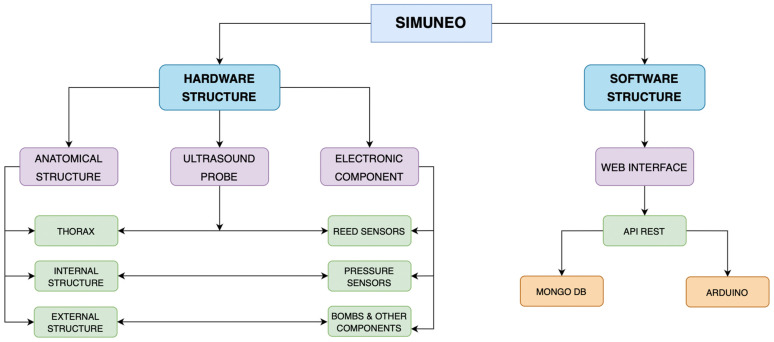
Scheme of SIMUNEO.

**Figure 2 sensors-23-05966-f002:**
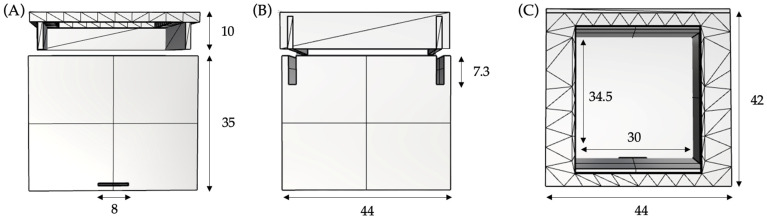
Internal box views and main measurements in millimetres. (**A**) Front view; (**B**) right view; (**C**) upper view.

**Figure 3 sensors-23-05966-f003:**
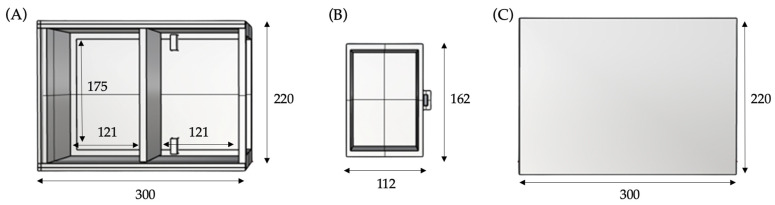
Upper view of the external structure and main measurements in millimetres. (**A**) Fixed box; (**B**) mobile box; (**C**) lid of the external structure.

**Figure 4 sensors-23-05966-f004:**
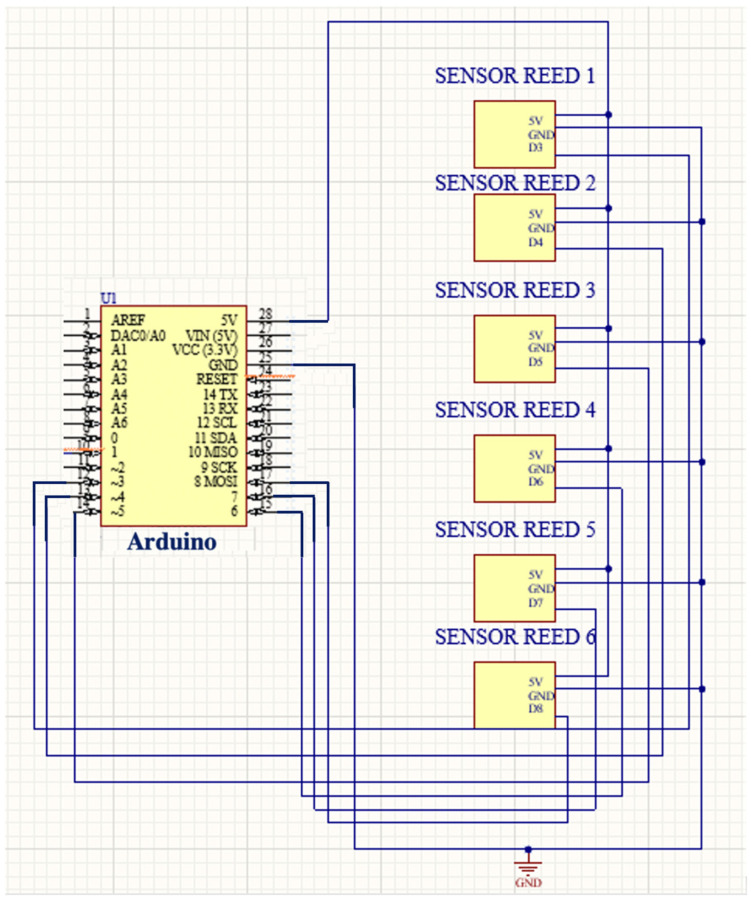
Reed sensor circuit: ultrasound simulation.

**Figure 5 sensors-23-05966-f005:**
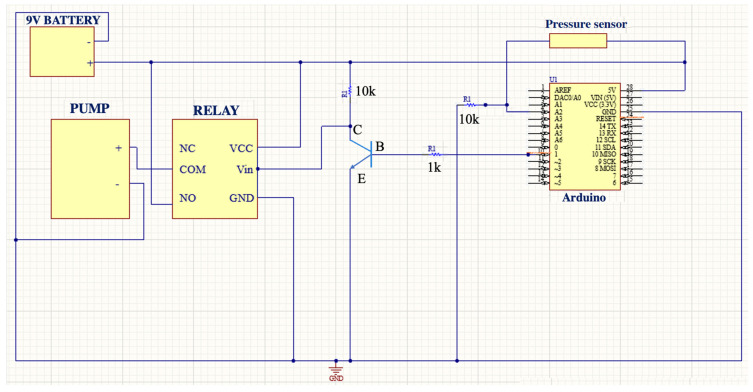
Pressure sensor circuit: decompression of pneumothorax and pleural effusion simulation.

**Figure 6 sensors-23-05966-f006:**
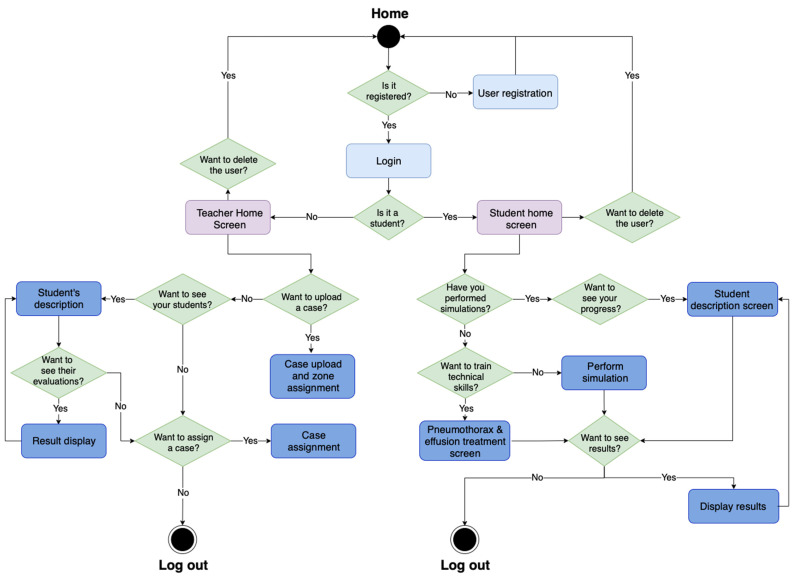
State diagram of the application.

**Figure 7 sensors-23-05966-f007:**
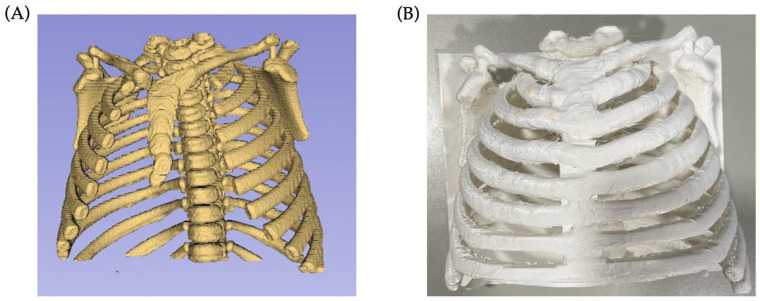
Progress of thorax reconstruction. (**A**) Initial segmentation of the thoracic skeleton from CT scans using 3D Slicer; (**B**) printed thorax of the final design.

**Figure 8 sensors-23-05966-f008:**
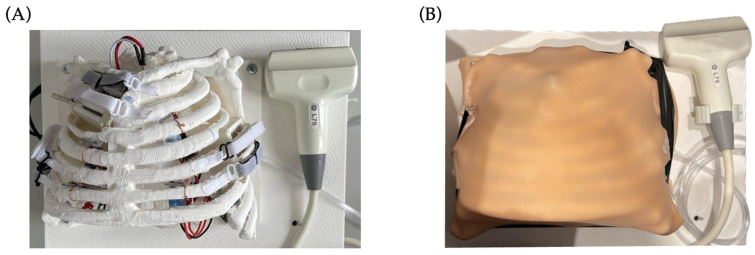
Final hardware composition. (**A**) Hardware structure without skin; (**B**) hardware structure with silicone simulating skin.

**Figure 9 sensors-23-05966-f009:**
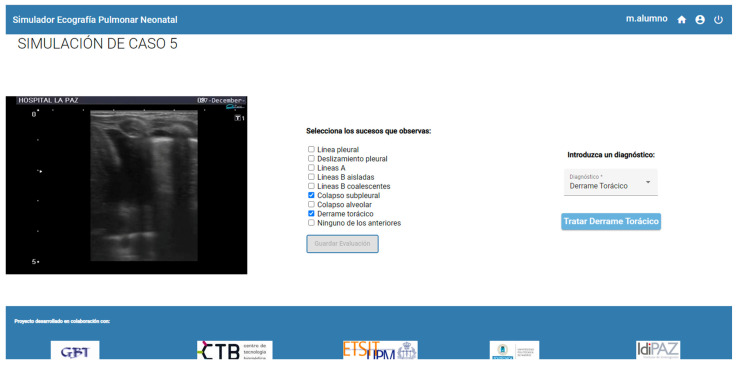
Simulation of ultrasound scanning: web application screen.

**Figure 10 sensors-23-05966-f010:**
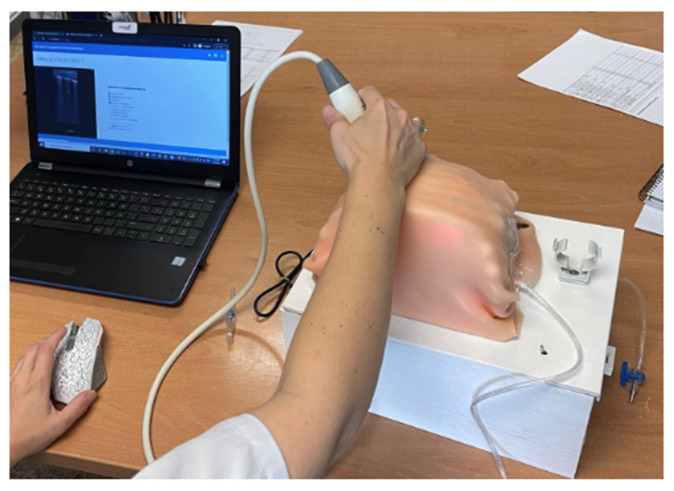
Simulation of ultrasound scanning: hardware usage.

**Figure 11 sensors-23-05966-f011:**
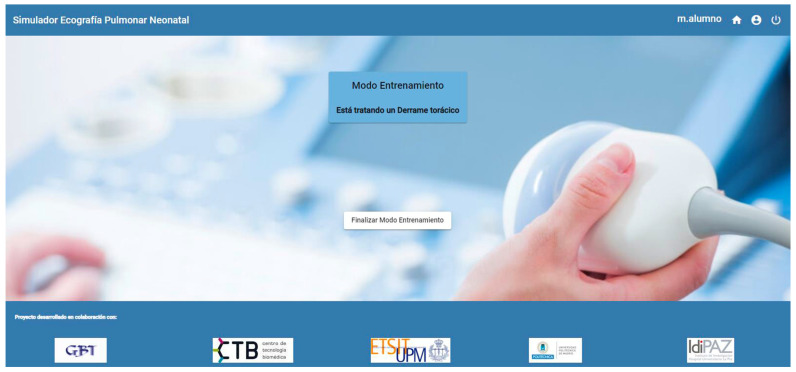
Simulation of a thoracic effusion: web application screen.

**Figure 12 sensors-23-05966-f012:**
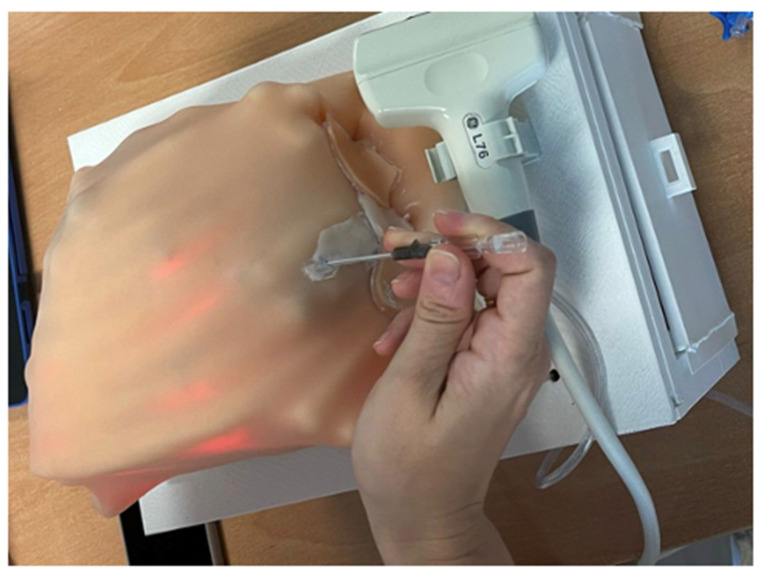
Simulation of a thoracic effusion: hardware usage.

**Table 1 sensors-23-05966-t001:** Commercial simulators and their collected information.

Simulators	Pneumothorax	Effusion	Lung Ultrasound	Control Software	Metrics and Evaluation
Micro-Preemie Nasco—LF01280	Yes	Yes			SM
PEDI^®^ Recién Nacido S320	Yes				SM
TruBaby X	Yes				SM
SUPER TORY S2220	Yes	Yes		Yes	OM
STAT Baby avanzado Nasco—101-8010	Yes	Yes		Yes	OM
NENASIM	Yes	Yes		Yes	OM
CAE BabySim	Yes	Yes		Yes	OM
CAE Luna	Yes	Yes		Yes	OM
LuSi	Yes			Yes	OM
BABYWORKS			Yes	Yes	OM
Echocom|Neo			Yes	Yes	OM
POCUSNEO			Yes	Yes	OM
diSplay U/S			Yes	Yes	OM

Subjective metrics (SMs) and objective metrics (OMs).

**Table 2 sensors-23-05966-t002:** Student usability questionnaire.

Questions	Student 1	Student 2	Student 3	Student 4	Student 5	Mean
I think I would like to use SIMUNEO frequently	5	5	5	5	5	5
SIMUNEO is easy to use	5	5	5	5	5	5
I do not think I would need the help of an expert to use SIMUNEO	3	5	3	4	5	3.89
I found the different parts of the system to be well integrated	3	5	5	4	4	4.13
I found the system to be very convenient for the realisation of the techniques	4	5	4	4	5	4.37
I felt very confident in the handling of the system	4	5	3	5	5	4.32
No extra knowledge is required to learn how to operate the system	4	5	3	3	5	3.90
I found the materials used to be realistic	4	5	5	4	4	4.37
The sensation experienced during the puncture was realistic	3	5	5	4	4	4.12
I found it easy to replace the water and batteries	4	5	3	4	4	3.95
I believe that the simulator will allow me to gain confidence and knowledge for my profession	5	5	5	5	5	5
I believe that SIMUNEO ensures patient safety	5	5	5	5	5	5

## Data Availability

The data presented in this study are available on request from the corresponding author.

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
