# Peer review of "SIMUNEO: Control and Monitoring System for Lung Ultrasound Examination and Treatment of Neonatal Pneumothorax and Thoracic Effusion"

_sensors, 2023, doi:10.3390/s23135966_

Round 1

Reviewer 1 Report

Dear Authors,

Overall, this manuscript is interesting, shows a significant amount of work, and describes a newly designed novel simulator for an important training niche. However, some reasonable effort will be required to bring it to a publishable standard.

The Introduction starts with the medical rationale for the project; however, the level of medical knowledge/writing is insufficient and requires medical input. Please consider inviting a medical collaborator or co-author to address the medical components of this introduction.

The Materials and Methods section describes the software, hardware, and system architecture in detail; however, the text would benefit greatly by making it more concise. Consider moving some of the Figures to a supplementary file.

The research component of this manuscript is a very short section entitled ‘Validation design’ which needs more detail. It should include the research questions, information about the participants, and the location and date(s) on which the research was done. Importantly, it should describe how the registrars (participants) were recruited and whether they consented to volunteer. In particular, please state your institutional ethical approval. Further details regarding the questionnaire are needed. The Discussion section overstates the findings, given the small number of evaluators, as does the Conclusion.

Further specific comments are provided below which I hope are helpful.

Abstract:

Please review the abstract in light of the manuscript preparation section of the instructions for authors.

Line 21 – you say that patient safety has been identified in the educational field as a key need, however is this not a key need in the medical field? Please review the sentence.

Introduction:

Page 1:

Line 40 – please check the plural of pneumothorax in this and all other instances.

Line 42 – check grammar

Line 44 – please provide a reference for the sentence which concludes on this line.

Page 2:

Line 51 – the paragraph which starts ‘the same way… ', needs review by a neonatal medical specialist for clinical relevance.

Line 62 – paragraph starting ‘in the 90s…’, two references are provided for historical information. As these two references are from 2015 and 2020, it appears these references are secondary in nature and the reference list should reflect this. Alternatively, provide original citations. In addition, please review this paragraph for grammar. Also, it is claimed at this technique is not commonly used despite several benefits, however I don't believe this to be true since the Covid pandemic.

Line 72 – paragraph starting ‘in order to…’. Please review this paragraph to make it more concise. Clinical simulation is an excepted technique, and much of this paragraph is unnecessary, you could also consider deleting it.

Line 87 – ‘preserving patient’ can be deleted.

Page 3:

Line 107 – please refer to tables and figures in the text prior to them appearing. In this instance you could just move the paragraph that appears below table 2, in front of it.

Materials and methods:

Page 4:

Line 140 – creating, rather than create.

Line 147 –please review the paragraph starting with ‘in addition…’ for grammar.

Page 5:

Line 161 – did you mean design rather than designed?

Line 162 - please review the sentence starting ‘another detail’ for readability.

Page 7:

Line 218 – paragraph starting with ‘the student role…’, please review for grammar.

Line 236 – sentence beginning ‘in the first…’ does not make sense.

Line 247 - sentence beginning ‘this data…’ could be removed to reduce repetition.

Page 9:

Line 275 – Validation design:

This appears to be the experimental component of the research. As such, please include your research questions and information about your participants, as well as consent to participate (in particular, institutional ethical approval), and further details regarding the questionnaire. Typically, users are included in the design stage prior to validation, please comment on the participation of users or a justification for not including users.

Page 10:

Line 311 - please review the sentence starting with ‘the precise…’. for readability.

Page 11:

Figure 9 – could figure 9 be revised, such that the text within the figure is in English?

Figure 10 – same comment as for Figure 9.

Line 340 – please review the use of the word ‘punctuations’ as I'm not sure what you mean.

Table 2 – please check the location of some of the scores within the table. Further information regarding the questionnaire would be useful for the reader.

Discussion:

Page 12:

Line 359 – it is claimed that the system resulted in ‘an intuitive, accessible, useful, and effective design’, however none of the questions asked the participants whether they found the equipment intuitive, accessible, useful, or effective. More nuance is required here to report the results accurately.

Line 362 – the paragraph starting ‘according to the results…’ discusses the sensations experienced by the testers. However, this is not present in the results for a reader to make sense of the claims, or come to their own conclusions.

Line 370 – as per the previous comment, the claim that the simulator ‘empowers’ clinical staff requires more evidence, as this was not obvious from the results and seems somewhat overstated.

Line 376 – sentence beginning with ‘the results show that…’ would be seen by many readers extrapolating beyond the reported results, e.g., the findings of the five residents. It is recommended to keep your claims aligned with the modest scope of your experimental design.

Page 13:

Line 386 – this paragraph appears to outline limitations. Please include further limitations such as the very small sample size of testers.

Conclusions:

Page 14:

Line 434 – is the simulator a promising solution in a ‘clinical’ field, or is it a solution for educational training?

Minor editing of English language required, see also the specific comments.

Author Response

We really appreciate the review done and comments, which will definitely help us to increase the quality of the publication. We have tried to address every comment in the best way. Please find bellow responses point by point, together with the new version of the manuscript with control changes. Our response follows (the reviewer’s comments are in italics and bold, page and line numbers refer to the revised manuscript with active change tracking)

  1. The Introduction starts with the medical rationale for the project; however, the level of medical knowledge/writing is insufficient and requires medical input. Please consider inviting a medical collaborator or co-author to address the medical components of this introduction.

Thank you so much for this comment. The medical collaborators of the project have reviewed and revised the introduction section concerning the presentation and resolution of diseases, as well as the section on clinical simulation. The introduction has been thoroughly reviewed and rewritten by the collaborating doctors involved in the project.

  1. The Materials and Methods section describes the software, hardware, and system architecture in detail; however, the text would benefit greatly by making it more concise. Consider moving some of the Figures to a supplementary file.

Thanks for the indication. The materials and methods section has undergone revisions in specific subsections to provide a more precise representation of its components. The following subsections in particular, have been modified to enhance clarity regarding their content:

  • 1 Hardware design: lines 155-159. The simulator hardware comprises the physical components of the system. This portion can be further divided into two distinct parts based on their purpose: the anatomical structure and the electronics. The complete anatomical structure, which houses a portion of the electronics, is covered with a silicone material that mimics human skin. This realistic simulation is essential for providing a lifelike experience during the training exercises.
  • 1.1 Anatomical structure: lines 190-194. Another noteworthy detail is the design of the hollow sliding cover. This design enables the needle to be inserted through the central hole, which leads to the electronic activation mechanism when it comes into contact with the pressure sensor. Additionally, it allows for the attachment of the ribs corresponding to the intercostal space, ensuring a secure fit of the box and facilitating the final assembly process
  • 2.1 Actors and use cases: lines 251-258. The student role is designed for medical students and resident physicians, either to learn the technique of lung ultrasound and decompression of pneumothorax and effusion from scratch in a real environment that ensures patient safety, allowing them to make mistakes as many times as necessary, or to train in these techniques at specific times during their training as desired. Students are supervised by an assigned teacher who oversees their progress. They can perform simulation cases assigned by their teacher, practice the pneumothorax and effusion technique on their own, or review their evaluations and track their progression
  • 4 Validation design: lines 319-336. For the initial presentation of the SIMUNEO prototype, a validation process is proposed involving one of the collaborating entities of the project, the Hospital Universitario de La Paz. The neonatology department of the hospital has been actively involved in the project, and it is suggested to conduct the validation with members from this department. As SIMUNEO aims to serve as a training tool in the clinical setting, the first validation of the system will involve five residents from the department. The reason for selecting residents as the initial group for validation in the medical field is because they are currently in their training period. This enables them to effectively compare the traditional learning method with SIMUNEO, allowing for a more objective evaluation of the system's realism and its similarities to their typical training. These residents, from different years of the neonatology service at Hospital Universitario La Paz in Madrid, have been recruited through the specialists of the department and have willingly chosen to participate in the validation process. As the recruitment took place within the neonatology department, the validation will also be conducted within this specific department of the hospital.

The validation that will be conducted consists of twelve statements that the residents will need to answer after completing a full simulation exercise. Additionally, short interviews will be conducted with the residents to gather their personal experiences and perspectives on the simulator.

  1. The research component of this manuscript is a very short section entitled ‘Validation design’ which needs more detail. It should include the research questions, information about the participants, and the location and date(s) on which the research was done. Importantly, it should describe how the registrars (participants) were recruited and whether they consented to volunteer. In particular, please state your institutional ethical approval.

Thank you so much for this comment. The validation design section has been carefully revised and clarified to provide a clear understanding of the participant group involved in the clinical validation, specifically the residents. Furthermore, the reason for selecting this particular group within the clinical context has been specified. The recruitment process for the residents and their voluntary participation have also been clearly explained. Additionally, this section has been enhanced by including the statements that form the basis of the validation process.

  1. Further details regarding the questionnaire are needed. The Discussion section overstates the findings, given the small number of evaluators, as does the Conclusion.

Thank you very much for your feedback. The discussion and conclusions sections have been revised to accurately reflect the actual findings of the simulator. Specifically, adjustments have been made to the previously overly optimistic conclusions, as they were not in line with the limited number of participants in the validation. Both sections now clarify that this small-scale validation serves as an initial assessment of the first developed prototype. The results obtained indicate the potential of the simulator as a training tool in the future, contingent upon implementing the suggested improvements and conducting further validation with a larger participant pool to obtain conclusive and objective outcomes.

  1. Abstract: Please review the abstract in light of the manuscript preparation section of the instructions for authors. Line 21 – you say that patient safety has been identified in the educational field as a key need, however is this not a key need in the medical field? Please review the sentence.

 Thank you so much for this comment. We have carefully reviewed the sentence and made revisions to better convey the message. The rewritten version emphasizes that the simulator aspires to serve as a valuable training tool in the clinical setting. It enables doctors in training to practice and enhance their skills, while building confidence, without having to directly practice on patients, thus indirectly contributing to ensuring patient safety, as trainee clinicians do not perform techniques on patients for the first time.

  1. Introduction:
    • Line 40 – please check the plural of pneumothorax in this and all other instances.

Thanks for the indication. We have corrected the error.

  • Line 42 – check grammar.

Thanks for the indication. We have we have revised the grammar and corrected errors.

  • Line 44 – please provide a reference for the sentence which concludes on this line.

Thank you so much for this comment. As a result of the revision conducted by clinical professionals, both the information and the references included in this section of the text have been modified.

  • Line 51 – the paragraph which starts ‘the same way… ', needs review by a neonatal medical specialist for clinical relevance.

Thank you so much for this comment. As a result of the revision conducted by clinical professionals the information has been reviewed and corrected.

  • Line 62 – paragraph starting ‘in the 90s…’, two references are provided for historical information. As these two references are from 2015 and 2020, it appears these references are secondary in nature and the reference list should reflect this. Alternatively, provide original citations. In addition, please review this paragraph for grammar. Also, it is claimed at this technique is not commonly used despite several benefits, however I don't believe this to be true since the Covid pandemic.

Thank you so much for this comment. This paragraph has been revised by a collaborating physician of the project in the following manner:

Over the past 20 years, LUS has been successfully used to diagnose and monitor lung diseases of the newborn. In addition, it played a crucial role in the disease follow up and facilitated lung diseases management procedures. Due to the sharp learning curve, ease of use, minimization of overall radiation exposure, the application of LUS has significantly increased or even completely replaced CTX in some neonatal intensive care units (NICU) across the world”.

The previous paragraph can be found in lines 67-72.

  • Line 72 – paragraph starting ‘in order to…’. Please review this paragraph to make it more concise. Clinical simulation is an excepted technique, and much of this paragraph is unnecessary, you could also consider deleting it.

Thanks for the indication. The paragraph has been reviewed by the physicians involved in the project, who have chosen to keep the information about clinical simulation. They consider it relevant for comprehending this aspect of the project and providing necessary context to the reader.

  • Line 87 – ‘preserving patient’ can be deleted.

Thank you so much for this comment. We have deleted that in the text.

  • Line 107 – please refer to tables and figures in the text prior to them appearing. In this instance you could just move the paragraph that appears below table 2, in front of it.

Thank you so much for this comment. We have changed the order of the paragraph and placed it above the table as you suggest.

  1. Materials and methods:
    • Line 140 – creating, rather than create.

Thank you so much for this comment. We have changed the word as you suggest.

  • Line 147 –please review the paragraph starting with ‘in addition…’ for grammar.

Many thanks for your appreciation. We have we have revised the grammar and corrected errors.

  • Line 161 – did you mean design rather than designed?

Thanks for the indication. We have changed the word to design as you mention.

  • Line 162 - please review the sentence starting ‘another detail’ for readability.

Thank you so much for this comment. We have changed that long sentence for shorter sentences so that the reader can understand the content well and the reading becomes more pleasant.

  • Line 218 – paragraph starting with ‘the student role…’, please review for grammar.

Many thanks for your appreciation. We have we have revised the grammar and corrected errors.

  • Line 236 – sentence beginning ‘in the first…’ does not make sense.

Thank you so much for this comment. We have revised the sentence and replaced it with a shorter and more concise one: “The user entity stores all the data of the users who make use of the web application”.

  • Line 247 - sentence beginning ‘this data…’ could be removed to reduce repetition.

Many thanks for your suggestion, information was indeed repeated, so we have removed the sentence as you mention.

  • Line 275 – Validation design:This appears to be the experimental component of the research. As such, please include your research questions and information about your participants, as well as consent to participate (in particular, institutional ethical approval), and further details regarding the questionnaire. Typically, users are included in the design stage prior to validation, please comment on the participation of users or a justification for not including users.

Many thanks for your appreciation. The validation design section underwent a thorough review and enhancements to ensure a comprehensive understanding of the participant group engaged in the clinical validation, particularly the residents. Moreover, the rationale behind selecting this specific group within the clinical context was clearly outlined. The recruitment process of the residents and their voluntary participation were also elucidated. Furthermore, this section was strengthened by incorporating the statements that serve as the foundation of the validation process.

  • Line 311 - please review the sentence starting with ‘the precise…’. for readability.

Thank you so much for this comment. We have changed that long sentence for shorter sentences so that the reader can understand the content well and the reading becomes more pleasant.

  • Figure 9 – could figure 9 be revised, such that the text within the figure is in English? Figure 10 – same comment as for Figure 9.

Many thanks for your appreciation. The text in figures 9 and 10 cannot be changed to another language in this first prototype of SIMUNEO, as it has been developed in collaboration with the Hospital Universitario La Paz, with the aim of implementing it in their learning courses. Teaching at this hospital is carried out in Spanish, so in this first system there is no possibility of changing the language of the interface. However, thank you very much for your comment, we will consider it for future lines of development of SIMUNEO.

  • Line 340 – please review the use of the word ‘punctuations’ as I'm not sure what you mean.

Thank you so much for this comment. The word has been changed to scores.

  • Table 2 – please check the location of some of the scores within the table. Further information regarding the questionnaire would be useful for the reader.

 Thank you so much for this comment. The layout of the table has been corrected to show readable data. Moreover, the validation design section now incorporates the information regarding the questionnaire, including the statements it comprises and the scoring methodology employed. In response to earlier feedback, a paragraph describing the brief interviews conducted with the validation participants has been added, which were initially overlooked. These interviews provide valuable personal perspectives from the participating residents in the initial validation of the system.

  1. Discussion:
    • Line 359 – it is claimed that the system resulted in ‘an intuitive, accessible, useful, and effective design’, however none of the questions asked the participants whether they found the equipment intuitive, accessible, useful, or effective. More nuance is required here to report the results accurately.

Thank you so much for this comment. We have tried to improve the explanation of the mentioned part by adding as justification the answers related to the ease of use, the requirement for additional knowledge or assistance from a professional, and the confidence level when using the simulator. Given that the score shown in table 2 is higher than the average score of 3, we have justified that SIMUNEO aims to be an intuitive, accessible, useful, and effective system.

  • Line 362 – the paragraph starting ‘according to the results…’ discusses the sensations experienced by the testers. However, this is not present in the results for a reader to make sense of the claims, or come to their own conclusions.

Thank you so much for this indication. The discussion section has undergone significant modifications to effectively communicate the experience of developing this initial SIMUNEO prototype and the results obtained from the first validation. Although these results are not definitive, they provide valuable guidance, positioning the project as a promising tool for further development.

  • Line 370 – as per the previous comment, the claim that the simulator ‘empowers’ clinical staff requires more evidence, as this was not obvious from the results and seems somewhat overstated.

Thank you so much for this indication. We have made revisions to this section in complete agreement with the feedback you provided regarding the discussion. Specifically, we have modified this particular part by replacing the previous text with a more realistic and result-oriented fragment.

“Overall, the residents' perception of the system is positive, as they view SIMUNEO as a beneficial tool that allows them to train, make mistakes, learn from them, and enhance their techniques and self-confidence”.

  • Line 376 – sentence beginning with ‘the results show that…’ would be seen by many readers extrapolating beyond the reported results, e.g., the findings of the five residents. It is recommended to keep your claims aligned with the modest scope of your experimental design.

Many thanks for your suggestion, we have made revisions to this section in complete agreement with the feedback you provided regarding the discussion.

  • Line 386 – this paragraph appears to outline limitations. Please include further limitations such as the very small sample size of testers.

 Thank you so much for this comment. We have added:

“It is important to exercise caution in interpreting them due to the small sample size, which may not be fully representative. Therefore, it is essential to conduct additional validations with a larger number of participants to further substantiate these findings.”

  1. Conclusions: Line 434 – is the simulator a promising solution in a ‘clinical’ field, or is it a solution for educational training?

Thanks for the indication. Thanks to the feedback provided, we are aware that we have not made the purpose of the simulator clear. What we want to convey is that the simulator aspires to serve as a valuable training tool in the clinical setting. It enables doctors in training to practice and enhance their skills, while building confidence, without having to directly practice on patients, thus indirectly contributing to ensuring patient safety, as trainee clinicians do not perform techniques on patients for the first time. The conclusions section has been modified in accordance with the aforementioned points.

Reviewer 2 Report

I have the following comments for the authors to help them enhance their manuscript:

1. In the introduction section, technically, which kind of scientific problem the authors try to solve? 

2. What's the major differences of SIMUNEO system compared to the simulators shown in Table 1? 

3. A detailed explanation of SIMUNEO system suggests to being added in section 2. The introduction of the function of each component is also necessary. 

4. A comparison study between SIMUNEO and other simulators suggests to being included in the results section. 

Author Response

We really appreciate the review done and comments, which will definitely help us to increase the quality of the publication. We have tried to address every comment in the best way. Please find bellow responses point by point, together with the new version of the manuscript with control changes. Our response follows (the reviewer’s comments are in italics and bold, page and line numbers refer to the revised manuscript with active change tracking)

I have the following comments for the authors to help them enhance their manuscript:

  1. In the introduction section, technically, which kind of scientific problem the authors try to solve? 

Thank you so much for this comment.  The problem is outlined in the following paragraph, which has been included in the abstract:

“Training on real patients is a critical aspect of the learning and growth of doctors in training. However, this essential step in the educational process for clinicians can potentially compromise patient safety, as they may not be adequately prepared to handle real-life situations independently.”

What we want to convey is that the simulator aspires to serve as a valuable training tool in the clinical setting. It enables doctors in training to practice and enhance their skills, while building confidence, without having to directly practice on patients, thus indirectly contributing to ensuring patient safety, as trainee clinicians do not perform techniques on patients for the first time. The conclusions section has been modified in accordance with the aforementioned points.

  1. What's the major differences of SIMUNEO system compared to the simulators shown in Table 1? 

Thank you so much for this comment. The most significant distinction between the simulators presented in Table 1 and SIMUNEO is that the proposed system enables the execution of all three techniques mentioned in the table, a capability none of the other simulators in the table possess concurrently. Moreover, SIMUNEO is accompanied by dedicated software and metrics to facilitate feedback. This entire section has been revised to enhance clarity regarding the fundamental differences between the systems featured in the table and how SIMUNEO sets itself apart from them.

  1. A detailed explanation of SIMUNEO system suggests to being added in section 2. The introduction of the function of each component is also necessary. 

Thanks for the indication. The materials and methods section has undergone revisions in specific subsections to provide a more precise representation of its components. The following subsections in particular, have been modified to enhance clarity regarding their content:

  • 1 Hardware design: lines 155-159. The simulator hardware comprises the physical components of the system. This portion can be further divided into two distinct parts based on their purpose: the anatomical structure and the electronics. The complete anatomical structure, which houses a portion of the electronics, is covered with a silicone material that mimics human skin. This realistic simulation is essential for providing a lifelike experience during the training exercises.
  • 1.1 Anatomical structure: lines 190-194. Another noteworthy detail is the design of the hollow sliding cover. This design enables the needle to be inserted through the central hole, which leads to the electronic activation mechanism when it comes into contact with the pressure sensor. Additionally, it allows for the attachment of the ribs corresponding to the intercostal space, ensuring a secure fit of the box and facilitating the final assembly process
  • 2.1 Actors and use cases: lines 251-258. The student role is designed for medical students and resident physicians, either to learn the technique of lung ultrasound and decompression of pneumothorax and effusion from scratch in a real environment that ensures patient safety, allowing them to make mistakes as many times as necessary, or to train in these techniques at specific times during their training as desired. Students are supervised by an assigned teacher who oversees their progress. They can perform simulation cases assigned by their teacher, practice the pneumothorax and effusion technique on their own, or review their evaluations and track their progression
  • 4 Validation design: lines 319-336. For the initial presentation of the SIMUNEO prototype, a validation process is proposed involving one of the collaborating entities of the project, the Hospital Universitario de La Paz. The neonatology department of the hospital has been actively involved in the project, and it is suggested to conduct the validation with members from this department. As SIMUNEO aims to serve as a training tool in the clinical setting, the first validation of the system will involve five residents from the department. The reason for selecting residents as the initial group for validation in the medical field is because they are currently in their training period. This enables them to effectively compare the traditional learning method with SIMUNEO, allowing for a more objective evaluation of the system's realism and its similarities to their typical training. These residents, from different years of the neonatology service at Hospital Universitario La Paz in Madrid, have been recruited through the specialists of the department and have willingly chosen to participate in the validation process. As the recruitment took place within the neonatology department, the validation will also be conducted within this specific department of the hospital.

The validation that will be conducted consists of twelve statements that the residents will need to answer after completing a full simulation exercise. Additionally, short interviews will be conducted with the residents to gather their personal experiences and perspectives on the simulator.

Additionally, in each section that comprises the system, both hardware and software, a brief introduction can be found.

  1. A comparison study between SIMUNEO and other simulators suggests to being included in the results section. 

Thank you so much for this comment. This comparison can be found in the discussion section. This section has also been improved for further clarification.

Round 2

Reviewer 1 Report

Dear Authors, thank you for addressing the concerns raised in the previous version so thoroughly.

There are a number of small grammatical corrections to be reviewed.

Reviewer 2 Report

All of my comments have been addressed by the authors. I do not have further technical comments for the authors.